# Morphological and Molecular Characterization of Some *Alternaria* Species Isolated from Tomato Fruits Concerning Mycotoxin Production and Polyketide Synthase Genes

**DOI:** 10.3390/plants11091168

**Published:** 2022-04-26

**Authors:** Abdelrahman Saleem, Amany A. El-Shahir

**Affiliations:** Department of Botany and Microbiology, Faculty of Science, South Valley University, Qena 83523, Egypt; asaleem@hotmail.com or

**Keywords:** *Alternaria*, internal transcribed spacer, mycotoxins, HPLC, *pksH* & *pksJ* genes

## Abstract

Tomatoes (*Lycopersicon esculentum*) are one of the main crops grown in Egypt. The fungal black spot illness of fruits is usually associated with the secretion of mycotoxin by *Alternaria* toxigenic species. Twenty *Alternaria* isolates were isolated from infected tomatoes fruits by baiting technique, morphologically identified to species level, and confirmed using Internal Transcribed Spacer (ITS) gene sequencing. ITS gene sequencing of fragments obtained 547, 547, 542, 554, and 547 bp for *A. alternata*, *A. brassicicola*, *A. citri*, *A. radicina*, and *A. tenuissima*, respectively. *Alternaria* species were investigated for mycotoxin production using the high-performance liquid chromatography (HPLC) technique. The data from the HPLC analysis showed that the mycotoxins were determined in four out of five *Alternaria* species, with the incidence ranging from 0.89–9.85 µg/mL of fungal extract at different retention times. *Alternaria alternata* was the most active species and produced three types of toxins. Polyketide synthase genes (*pksH* and *pksJ*) which are involved in the *Alternaria* toxin’s biosynthesis were also amplified from the DNA of *Alternaria* species.

## 1. Introduction

*Alternaria* is a fungal genus with various saprophytic and pathogenic species found in nature. They could infect many crops in the field and throughout the postharvest period, resulting in losses due to fruit and vegetable rot [1,2,3,4]. The most abundant *Alternaria* species comprise *A. alternata*, *A. arborescens*, *A. brassicae*, *A. brassicicola*, *A. infectoria*, *A. radicina*, and *A. tenuissima*. They infect a wide variety of economically important plants, involving cereals, tomatoes, apples, grapes, oil crops, sunflower seeds, oranges, lemons, melons, cucumbers, cauliflower, peppers, and tangerines before harvest and under storage conditions [5,6,7]. Based on morphological and phylogenetic characteristics, the genus was recently divided into 26 sections [8], which contained several plant pathogens [9,10,11]. The growth of *Alternaria* on plants is usually associated with the secretion of mycotoxins. Bessadat et al. [12] studied the tomato early blight of *Alternaria* in Algeria’s northwest. *Alternaria* was recovered from 80% and 50% of diseased plants and total fungi, respectively. Morphological and molecular studies showed that small-spore isolates were common in most locations surveyed and reached above 80% of total *Alternaria* isolates. Regarding their sporulation patterns, they were identified as *A. alternata* and *A. tenuissima*. Large-spore isolates were recognized as *A. linariae*, *A. solani*, and *A. grandis* and were detected in all samples with 33.8%, 6.3%, and 1.3% of total *Alternaria*, respectively. Mohammadi and Bahramikia [13] studied the molecular identity and genetic differences of some *Alternaria* species recovered from black-spot tomatoes by internal transcribed spacer (ITS1) sequencing. Regarding the ITS1 sequencing, all isolates were *A. alternata*. The ITS1 sequencing differentiated the *Alternaria* spp. that had similar morphology. *A. alternata* contaminates human food and livestock by secretion of many mycotoxins [4,5,6]. Recently, Aung et al. [14] studied morphology and molecular description of small-spore *Alternaria* which cause the black spots of stored pear fruits grown in China. According to morphological characterization, the tested *Alternaria* was similar to the species *Alternaria limoniasperae*, *A. perangusta*, *A. interrupta*, and *A. turkisafria*. A phylogenetic analysis created by several sequences datasets including ITS and Endopolygalacturonase (EndoPg) genes showed that the tested *Alternaria* was one of the *Alternaria alternata* complex group.

The European Food Safety Authority (EFSA) [15] has identified several *Alternaria* secondary metabolites as potentially hazardous to human health. The studied *Alternaria* secondary metabolites belong to diverse chemical groups such as nitrogen-containing compounds (amide, cyclopeptides), steroids, terpenoids, pyranones, quinines, and phenolics. The perylenquinone derivatives, such as ATX I, ATX II, ATX III, Alterperylenol (ALTCH; synonym Alteichin), and stemphyltoxins (STE), are minor metabolites of *Alternaria* spp. but are considered to be very critical because of their mutagenic properties. However, *Alternaria* spp. produce a variety of other metabolites for which no reports are available due to the lack of pure substances. *Alternaria* species can produce more than 70 toxins, which play important roles in fungal pathogenicity and food safety since some of them are harmful to humans and animals [16]. *Alternaria* mycotoxins exhibit great structural divergence and are commonly divided into five groups: dibenzo-α-pyrones (alternariol, AOH; alternariol-monomethyl ether, AME), tetramic acid derivatives (tenuazonic acid, TA), perylene quinones (altertoxins I–III, ATX I–III), specific toxins produced by *Alternaria alternata* subspecies *lycopersici* (AAL-toxins), and miscellaneous structures (tentoxin, TEN) [2,4,5,17,18]. Among these mycotoxins are altenuene (ALT), alternariol (AOH), alternariol monomethyl ether (AME), tenuazonic acid (TA), and tentoxin (TEN), which are the most abundant in food and produced in relatively detectable amounts (Appendix A). Exposure to these toxins causes genotoxic, mutagenic, carcinogenic, and cytotoxic effects, as well as inhibition of enzymes activity, on both humans and animals. In addition, AOH and AME were able to induce DNA strand breaks and gene mutations in cultured human and animal cells. In the case of other *Alternaria* mycotoxins, the availability of labeled standards is still limited for analytical quantification [19,20,21]. Among *Alternaria* species that are well-known sources of these toxins are *A. alternata*, *A. brassicicola*, *A. cucumerina*, *A. dauci*, *A. raphani*, *A. solani*, and *A. tenuissima* [22,23,24]. *Alternaria* toxins such as ALT, AOH, AME, and TA have been detected in several plant crops including fruits, vegetables, and derived products such as carrots, olives, tomatoes, mandarin, oranges, peppers, melons, strawberry, and tomato products [2,20,25,26]. Recently, Masiello et al. [11] investigated the black point fungal disease of wheat and mycotoxigenic *Alternaria* spp. *Alternaria* spp. were recognized visually at species level. *Alternaria* strains phylogenetically collected in *Alternaria* section formed AOH, AME, and TA with amounts of 8064, 14,341, and 3683 µg/g, respectively. In contrast, eight *Alternaria* strains from the *Infectoriae* section produced a small amount of mycotoxin. Moreover, *A. alternata* fungus causes fungal allergies in humans and animals [27,28,29].

Polyketides, terpenes, and alkaloids are the prevalent pathways of fungal secondary metabolism [30]. AOH is biosynthesis from the polyketide pathway, a predominant route for the secretion of various fungal metabolites [31,32]. The main step in the biosynthesis of mycotoxins and other fungal metabolites is polyketide synthases (*pks*). Fungal polyketides are secreted by multi-domains. *Pks* domains include a starter unit ACP transacylase (SAT), b-ketoacyl synthase (KS), acyl transferase (AT), product template (PT), acyl carrier protein (ACP), and claisen-cyclase/thiolesterase (CLC/TE). In terms of domain structure, there are three types of fungal *pks*: highly reducing (HR), partially reducing (PR), and non-reducing (NR) [33]. In this respect, Saha et al. [18] supposed that the *pks* is the essential enzyme for *Alternaria* toxins biosynthesis. The genomic sequence of *A. alternata* includes ten genes that are believed to code for *pks*. *PksH* and *pksJ* genes are essential for enzymes secretion associated with the production of toxins by *Alternaria*. The proteins of these genes are portended to be 2222 and 2821 amino acids in length, respectively. In this respect, Zghair et al. [34] detected the *pksJ* gene that was responsible for AOH production from *Alternaria alternata*-infected tomato by early blight disease in Karbala city, Iraq. They detected AOH production by TLC and correlated it to the *pksJ* gene. They showed that 23 out of 24 isolates produced the AOH toxin. The target *pksJ* gene was amplified and produced bands that have 514 bp for all isolates that produced the toxin, except isolate No. 2 which did not produce AOH.

This article aimed to identify some *Alternaria* species which cause the black spot illness of tomato fruits according to morphological description and by ITS gene sequencing. In addition to the investigation of the ability of these species for *Alternaria* toxins production using the high-performance liquid chromatography (HPLC), we also used amplification of *pksH* and *pksJ* genes that were involved in the biosynthesis of *Alternaria* toxins.

## 2. Materials and Methods

### 2.1. Collection of Tomato Samples

In the summer of 2019, twenty samples of black spots symptoms tomato (*Lycopersicon esculentum*) fruits (cultivated plant) were collected randomly from various markets in Qena city (26°10′00″ N 32°43′00″ E), in Upper Egypt. The samples were collected in sanitized polyethylene bags and transported to a microbiology lab for fungal isolation (Appendix A).

### 2.2. Isolation and Morphological Description of Alternaria

Potato dextrose agar (PDA) medium which contained “200.0 g potato; 20.0 g dextrose; 15.0 g agar (Merck, Darmstadt, Germany), and chloramphenicol (0.05 g/L) as a bacteriostatic agent was used for the isolation of fungi.

Tomato fruits with black spots were used for isolation of fungi by baiting method as reported by Pitt and Hocking [35]. Tomato fruits were disinfected for 1 min with 1% sodium hypochlorite, rinsed three times with sterilized distilled water, and dried with filter papers before being sliced into equal segments (about 1 cm each). On PDA-medium-coated plates, five segments of spot samples were inserted. For each sample, five plates were created. The cultures were stored for 7 days at 25 °C. Using a light microscope (40 magnification), the developing *Alternaria* spp. were studied and morphologically characterized based on macro- and microscopic features such as conidia color, size, septation, conidiophore branching, and catenulate conidia diameters. The morphological identity of *Alternaria* spp. was performed according to the following references [36,37].

### 2.3. DNA Extraction of Alternaria Species

*Alternaria* cultures grown on the PDA at 25 °C for 7 days were employed for DNA extraction according to QIAamp DNeasy Plant Mini kit instructions. A total of 100 mg of fungal mycelia was frozen at −80 °C for 24 h for later processing. Fungal material and a tungsten carbide beads were added to a 2 mL safe-lock tube. Totals of 400 μL Buffer AP1 and 4 μL RNase A stock solution (100 mg/mL) were added. Tubes were placed into the adaptor sets, which are fixed into the clamps of the tissue Lyser. Disruption was performed in two 1–2 min high-speed (20–30 Hz) shaking steps. The mixture was incubated for 10 min at 65 °C and mixed 2 or 3 times during incubation by inverting the tube. A total of 130 μL Buffer P3 was added to the lysate, mixed, and incubated for 5 min on ice. The lysate was centrifuged for 5 min at 14,000 rpm. The lysate was pipetted into the QIAshredder Mini spin column (lilac), placed in a 2 mL collection tube, and centrifuged for 2 min at 14,000 rpm. The flow-through fraction was transferred into a new tube without disturbing the cell-debris pellet. A total of 1.5 volume of Buffer AW1 was added to the cleared lysate and mixed by pipetting. A total of 650 μL of the mixture (including any precipitate that was formed) was pipetted into the DNeasy Mini spin column, placed in a 2 mL collection tube, and centrifuged for 1 min at 8000 rpm, and the flow-through was discarded. The DNeasy Mini spin column was placed into a new 2 mL collection tube. A total of 500 μL Buffer AW2 was added and centrifuged for 1 min at 8000 rpm, and the flow-through was discarded. A total of 500 μL Buffer AW2 was added to the DNeasy Mini spin column and centrifuged for 2 min at 14,000 rpm to dry the membrane. The DNeasy Mini spin column was transferred to a 1.5 mL or 2 mL microcentrifuge tube, and 50 μL Buffer AE was directly pipetted onto the DNeasy membrane. It was incubated for 5 min at room temperature (15–25 °C) and then centrifuged for 1 min at 8000 rpm to elute [38,39].

### 2.4. Polymerase Chain Reaction, Amplification of 5.8S rDNA, and Gene Sequencing

The amplification of 5.8S rDNA was performed using the universal primer pair ITS1 (5′TCCGTAGGTGAACCTGCGG′3) and ITS4 (5′TCCTCCGCTTATTGATATGC′3) [40]. The PCR included 35 cycles, with an initial denaturation of 5 min at 94 °C, secondary denaturation for 30 s at 94 °C, primer annealing for 40 s at 56 °C, primer extension for 45 s at 72 °C, and a final extension for 45 s at 72 °C. The reaction was made in a quantity of 25 µL containing 6 µL DNA templates, 12.5 µL master mix, and 1 µL (20 p mol) of each primer. The PCR products were electrophoresed (for 1 h at 80 V) in 1.0% agarose gel in Tris-borate-EDTA buffer at pH 8. The gel was stained with ethidium bromide and was observed in a gel documentation system. The PCR products were sequenced by Elim Biopharmaceuticals Inc. (a biotechnology company in Hayward, California, CA, USA). PCR product purification and gene sequencing (bidirectional sequencing) was performed as follows: QIAquick PCR product extraction kit (Qiagen Inc., Valencia, CA, USA) was used for the purification of the PCR product directly. A BigDye Terminator V3.1 cycle sequencing kit (Perkin-Elmer, Foster City, CA, USA) was used for performing gene sequencing using an Applied Biosystems 3130 genetic analyzer (Hitachi, Ltd., Tokyo, Japan). Centrisep (spin column)*:* cat number *CS-901* was used for 100 reactions. A purified PCR product was sequenced in the forward and reverse directions on an Applied Biosystems 3130 automated DNA Sequencer ABI, 3130 (Thermo Fisher Scientific Inc., Waltham MA, USA). Using a ready-reaction BigDye Terminator V3.1 cycle sequencing kit (Perkin-Elmer/Applied Biosystems, Foster City, CA, USA), the nucleotide sequences procured by the local BLAST (http://blast.ncbi.nlm.nih.gov/Blast.cgi, accessed on 12 April 2022) were tested and emulated with identical sequences in the GenBank [13]. ITS sequences revealed in our investigation, as well as validated sequences of all species discovered within *Alternaria* were phylogenetically examined to determine the taxonomic status of the isolates using recent molecular classification criteria [8]. Sequences were aligned by MUSCLE and introduced to MEGA-X version 10.2.2 software [41] for phylogenetic analysis with the maximum likelihood method [42]. Analyses were performed with 500 bootstrap replications and the phylogenetic tree was constructed.

### 2.5. Amplification of pksH and pksJ Genes

The *pksH* and *pksJ* genes, which contributed to the secretion of *Alternaria* toxins within the *Alternaria* gene cluster, were amplified by the specific primer pairs for *pksH* (GTCAACCCTCTCACACCAAC)-(GACGCATCGCTTCAATAGCC) and *pksJ* (GTCCCAAATTCCTACCCTCAC)-(GATAGCCATCGAAAGCATTCCC) [24]. The reaction was made in a quantity of 25 µL containing 6 µL DNA templates, 12.5 µL master mix, and 1 µL (20 p mol) of each primer. The PCR included 35 cycles as follows: primary denaturation for 5 min at 94 °C, secondary denaturation for 30 s at 94 °C, primer annealing for 30 s at 62 °C, extension for 30 s at 72 °C, and last extension for 7 min at 72 °C for both primer pairs. PCR products were visualized under UV light in 1.0% agarose gel stained by ethidium bromide.

### 2.6. Toxin Production and Purification for HPLC Analysis

Twenty *Alternaria* isolates from five different species were grown in three replicates in a 500 mL flask containing 100 mL PDB. The cultures were incubated for 15 days at 25 °C. After that, Whatman No. 1 filter paper was used to filter the cultures. Each filtrate (200 mL) was mixed with 100 mL of a 93:7 *v*/*v* chloroform-methanol mixture. For 30 min, the mixture was shaken. Chloroform was collected using a separating funnel, dried with anhydrous sodium sulfate, and evaporated at 40 °C near dryness in a rotary evaporator. The residue was diluted in 2 mL methanol and stored at 4 °C until further investigation [25,43,44].

### 2.7. Chromatographic Analysis

A Hewlett-Packard HP1050 liquid chromatographer (Palo Alto, CA, USA) was used in conjunction with a Rheodyne sample valve with a 20 mL loop and an HP diode array detector (model 1050, Phoenix: eBay Inc. Hamilton Ave., San Jose, CA, USA and MacroSpectro software: SPECTRO Analytical Instruments GmbH Boschstr. 1047533 Kleve, Germany). Spherisorb ODS-2, 5 mm, 250 mm, was used as the analytical column (Phase Separations Ltd., Deeside, Chwyd, UK). The mobile phase was 0.7 mL/min of methanol/water (80:20) containing 300 mg ZnSO_4_·H_2_O/L. Chromatograms had a wavelength of 250 nm. To quantify toxins and related peak areas against concentration, a calibration curve was created. The peak identity was established by contrasting the spectrum of the standard with the positive peak of the sample after normalization.

## 3. Results

### 3.1. Morphological Description of Alternaria Species

*Alternaria* spp. were the most common species isolated from the infected tomato fruits (Appendix A). Twenty isolates of *Alternaria* were grouped into five species, depending on their macro- and microscopic description on the PDA medium. These species were *A. alternata*, *A. brassicicola*, *A. citri*, *A. radicina,* and *A. tenuissima*. Variable colony morphology differentiation was noticed on the PDA and in the microscopic examination characteristics, allowing for differentiation of the isolates into five morphotypes *Alternaria* species (Figure 1). *Alternaria* species were identified by using a binocular microscope (at 40× magnification) with the following standard manuals [36,37]. Identification of *Alternaria* isolates was carried out depending on some morphological characters, such as conidial dimensions and septation patterns measurements. A high diversity level of culture morphology and the conidia grown on the PDA were observed. *Alternaria* species differed morphologically in terms of conidial length and width, as well as the number of septa (transverse/longitudinal septa) (Table 1). The mean conidial length varied from 21.5–31.4, 30.6–32.8, 27.1–41.4, 24.4–34.8, and 28.7–36.0 µm for *A. alternata*, *A. brassicicola*, *A. citri*, *A. radicina*, and *A. tenuissima,* respectively. However, the average conidial width varied from 10.1–14.3, 15.3–16.2, 10.5–15.9, 10.6–15.8, and 11.0–12.4 for these species, respectively.

### 3.2. Molecular Characterization of Alternaria Species Using ITS rDNA Gene Sequencing

Five *Alternaria* species recovered from tomatoes were selected for molecular identification using ITS1 and ITS4 rDNA gene sequencing. *Alternaria* species were designated as *A. alternata*, *A. brassicicola*, *A. citri*, *A. radicina,* and *A. tenuissima*. The fungal-specific universal primer pairs ITS1 (forward) and ITS4 (reverse) successfully amplified the ITS region from the DNA of all *Alternaria* spp. The fragment sizes determined by electrophoresis were about 600 bp (Figure 2A). ITS gene sequencing of produced fragments were 547, 547, 542, 554, and 547 bp for *A. alternata*, *A. brassicicola*, *A. citri*, *A. radicina*, and *A. tenuissima*, respectively. NCBI-BLAST had examined the obtained rDNA sequences. The morphological identification was validated by a BLAST search of the ITS rDNA sequences. Different *Alternaria* species have the closest match (97–98% similarity) in the NCBI GenBank database. The *Alternaria* species ITS rDNA sequences have been deposited in the NCBI GenBank database as *Alternaria alternata* mic21, *A. brassicicola* mic21, *A. citri* mic21, *A. radicina* mic21*,* and *A. tenuissima* mic21. The BLAST data validated the *Alternaria* species variability. The obtained phylogenetic tree of the five *Alternaria* species includes three branches. Branch I contained *A. alternata* and *A. tenuissima*, branch II included two species, *A. citri* and *A. brassicicola,* which were similar, and branch III contained *A. radicina* (Figure 2B). A comparison of these *Alternaria* species with the other *Alternaria* in the GenBank showed a similarity of 99–100% with several *Alternaria* species. For example, *A. alternata* was similar to some *Alternaria alternata* strains/isolates and closely separated from *A. alternata* isolate TJZYM, *A. alternata* Leonurus, and *A. alternata* strain TAA-05. However, *A. citri* was identical to certain strain/isolates and closely grouped to *A. citri* strain FJMYR6, *A. citri* isolates 1092, and *A. citri* strain 10.1. *A. brassicicola* was similar to *A. brassicicola* strain a1. *A. radicina* was separated individually near *A. radicina* strain A29, *A. radicina* AR018, *A. radicina* isolate 9/320, and *A. radicina* CBS.245.67 in the phylogenetic tree. However, *A. tenuissima* was similar to several *A. tenuissima* strains/isolates and closely separated with *A. tenuissima* strain Af/4/3, *A. tenuissima* WS11803, and *A. tenuissima* strain ZB11263554. Generally, the similarity among these species and the corresponding species in the GenBank was relatively high (99–100%) (Figure 3).

### 3.3. Amplification of pksH and pksJ Genes

Total DNA was extracted from five species of *Alternaria*: *A. alternata*, *A. brassicicola*, *A. citri*, *A. radicina*, and *A. tenuissima*. Polyketide synthase genes (*pksH* and *pksJ*) were successfully amplified from the DNA of the five *Alternaria* species. The amplified segments of *pksH* and *pksJ* genes were 290 and 260 bp, respectively (Figure 4).

### 3.4. Toxin Production and Purification for HPLC Analysis

The ability of five *Alternaria* species (*A. alternata*, *A. brassicicola*, *A. citri*, *A. radicina*, and *A. tenuissima*) to produce toxins was investigated. These toxins were altenuene, alternariol, alternariol methyl ether, and tenuazonic acid. The data of HPLC analysis showed that these toxins were found in four *Alternaria* species out of five, with the incidence ranging from 0.89 to 9.85 µg/mL of fungal extracts at different retention times. *Alternaria alternata* was the most active species and produced three types of toxins, ranging from 4.12–9.85 µg/mL of fungal extract. The highest value of toxins was recorded for Alternariol methyl ether (9.85 µg/mL) produced by *A. alternata,* followed by *A. radicina* (6.65 µg/mL). However, the other species produced the toxins with lower levels (0.89–5.12 µg/mL). On the other hand, *A. brassicicola* did not produce toxins. Compared with different types of toxins tested, alternariol methyl ether was the highest one (5.12–9.85 µg/mL of fungal extracts), followed by the alternariol toxin (0.89–5.12 µg/mL) and tenuazonic acid (1.51–4.12 µg/mL). Altenuene was produced only by *A. citri* (1.35 µg/mL) (Table 2, Figure 5).

## 4. Discussion

In agreement with our results, *Alternaria* spp. was the most predominant species isolated from different substrates in Qena governorate, Upper Egypt [5]. Morphological descriptions and measurements for different *Alternaria* species were variable and overlapping. Comparing our data with those in the previous works of literature for these species revealed that the obtained data are in agreement with those recorded by Meena et al. [2] for similar *Alternaria* species, including *A. alternata*, *A. brassicicola*, and *A. tenuissima*. The morphological description of species was based on Simmons’s [37] morphological features. Amplification and sequencing of the ITS gene confirmed the morphological identification of these species. *Alternaria* species under study were closely grouped with similar species in the GenBank. The phylogenetic analyses of *Alternaria* showed that the data are similar to those obtained by Ramezani et al. [7]. They differentiated *Alternaria* spp. of tomato early blight in Iran. The ITS region was amplified and sequenced to determine the molecular identity of *Alternaria* spp. The PCR products for six species were around 550 bp. The ITS sequences submitted to GenBank confirmed that the species are *A. alternata*, *A. tenuissima*, *A. arborescens*, *A. mimicula*, and *A. infectoria,* with 99% similarity. In this regard, Garganese et al. [24] examined 20 *Alternaria* isolates from the brown spot in tangerines (leaf and fruits) in the geographical areas of Italy and Spain. A two-gene phylogeny involving ITS and EndoPg sequences of isolates were used for identification. A total of 18 isolates were linked to *A. alternata,* and 2 isolates were grouped with the *A. arborescens*. Many clades revealed a range of *Alternaria* species morphotypes, including *A. alternata*, *A. tenuissima*, *A. citri*, *A. arborescens*, *A. limoniasperae*, and *A. toxicogenica*, based on a partial match between molecular clades and morphotypes. In Iran, Mohammadi and Bahramikia [13] identified *Alternaria* recovered from 60 black spots in tomatoes gathered from supermarkets and cultured on the PDA. The isolates identity was performed using morphological characteristics and ITS rDNA sequencing. The isolate gene fragment size was 360 bp. When sequences of gene fragments were compared, those of the NCBI had a similarity of 99–100% with *A. alternata*.

*Alternaria* produces more than 60 secondary metabolites. AOH and AME are among the most common mycotoxins in *A. alternata*. The toxicology and genetics of their biosynthesis have been partially studied. *Pks* are one of the important enzymes in the biosynthesis of these toxins. In a draft genome sequence of *A. alternata,* 10 *pks* encoding genes were detected. *PksJ* and *pksH* are among the essential genes correlated with the production of toxins [18]. In this study, we investigated the ability of different *Alternaria* species to produce various kinds of mycotoxins and the correlation between toxin production and their *pksH* and *pksJ* biosynthetic genes. The results indicated that the presence of these genes is widespread in all *Alternaria* species which are able to produce the toxins, and in the other species which do not have the ability to produce toxins, it is probable that other necessary genes may be missed in the isolates that couldn’t produce toxins. In this respect, Garganese et al. [24] investigated the toxigenic behavior of *Alternaria* isolates. They found that TeA was the most abundant mycotoxin. Isolates were synthesized AOH, AME, and ALT mycotoxins. AME production significantly varied among six morphotypes. Recently, Masiello et al. [11] studied the black point fungal disease of wheat and associated toxigenic *Alternaria* species. *Alternaria* toxins included AOH, AME, TA, and ALT. Ninety-two *Alternaria* strains were identified morphologically and with gene sequencing at species/section level and analyzed for their mycotoxin profiles using HPLC. Eighty-four strains, phylogenetically grouped in the *Alternaria* section, formed AOH, AME, and TA with values of 8064, 14,341, and 3683 µg/g, respectively. In contrast, eight *Alternaria* strains, related to *Infectoriae* section, had little or no ability to form mycotoxins. The same results were also obtained by Habib et al. [45]. They confirmed the ability of several *Alternaria* species/strains isolated from tomato plants to produce alternariol, alternariol methyl ether, altenuene, and tenuazonic acid. The values were 5634, 16,006, 5156, and 4507 mg/kg, respectively.

## 5. Conclusions

This article investigated the variance of five *Alternaria* species, molecular characterization, and toxigenicity. These species can infect tomato fruits and exhibit black spot disease. Infection of fruits by *Alternaria* pathogens is associated with the secretion of various mycotoxins which are harmful to human health. In Egypt, *Alternaria* has many pathogenic species that cause infections in a variety of economically important horticultural crops and vegetables. Molecular analysis and genetic variance of *Alternaria* species would be of great importance in plant breeding for disease resistance and plant protection. This study also confirmed the ability of these species to produce various types of mycotoxins (ALT, AOH, AME, and TA) compared with different *Alternaria* species which are deteriorating tomato fruits and reducing the quality and significance of tomato crop. The polyketide pathway is a valuable microbial secondary metabolism and natural product which is of great potential as a medicinal agent. The study of *pks* would contribute to biochemical studies and their applications in biotechnology.

## Figures and Tables

**Figure 1 plants-11-01168-f001:**
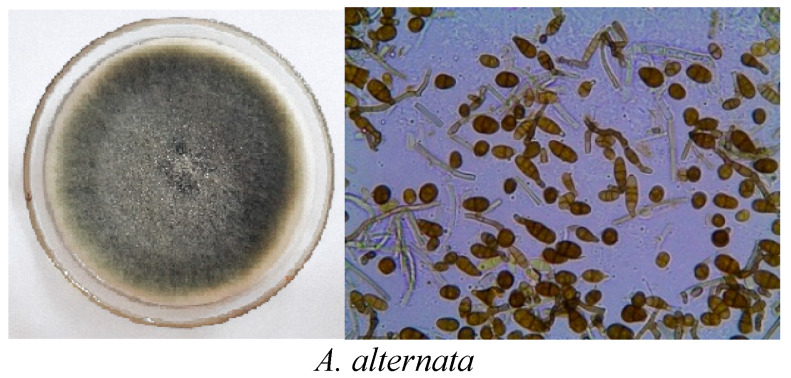
*Alternaria* species isolated from infected tomato fruits.

**Figure 2 plants-11-01168-f002:**
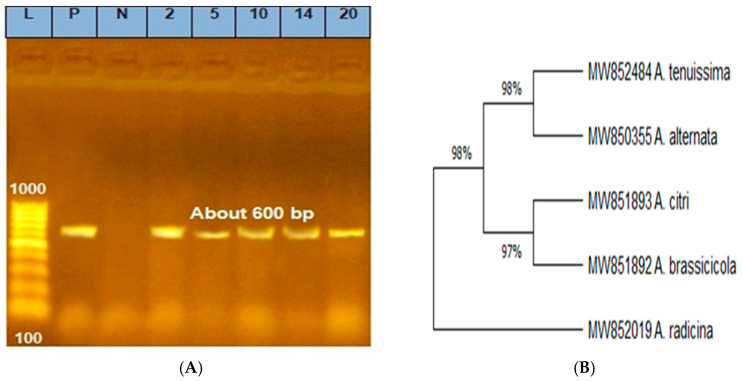
Agarose gel electrophoresis of PCR amplification of DNA products, PCR amplification of DNA products (~600 bp) using ITS1/ITS4 primer pair. L—ladder (100–1000 bp); P—positive control consists of a segment of DNA of known size (the same size as the target amplicon, shows that the primers have attached to the DNA strand); N—negative control: a sample without DNA, but contains all essential components of the amplification reaction show if contamination of the PCR experiment with foreign DNA has occurred; 2—*A. alternata*; 5—*A. citri*; 10—*A. brassicicola*; 14—*A. radicina*; 20—*A. tenuissima* (**A**) and phylogenetic tree generated from *Alternaria* species used in this study based on datasets of ITS rDNA gene sequences, the similarity of 97–98% among species (**B**).

**Figure 3 plants-11-01168-f003:**
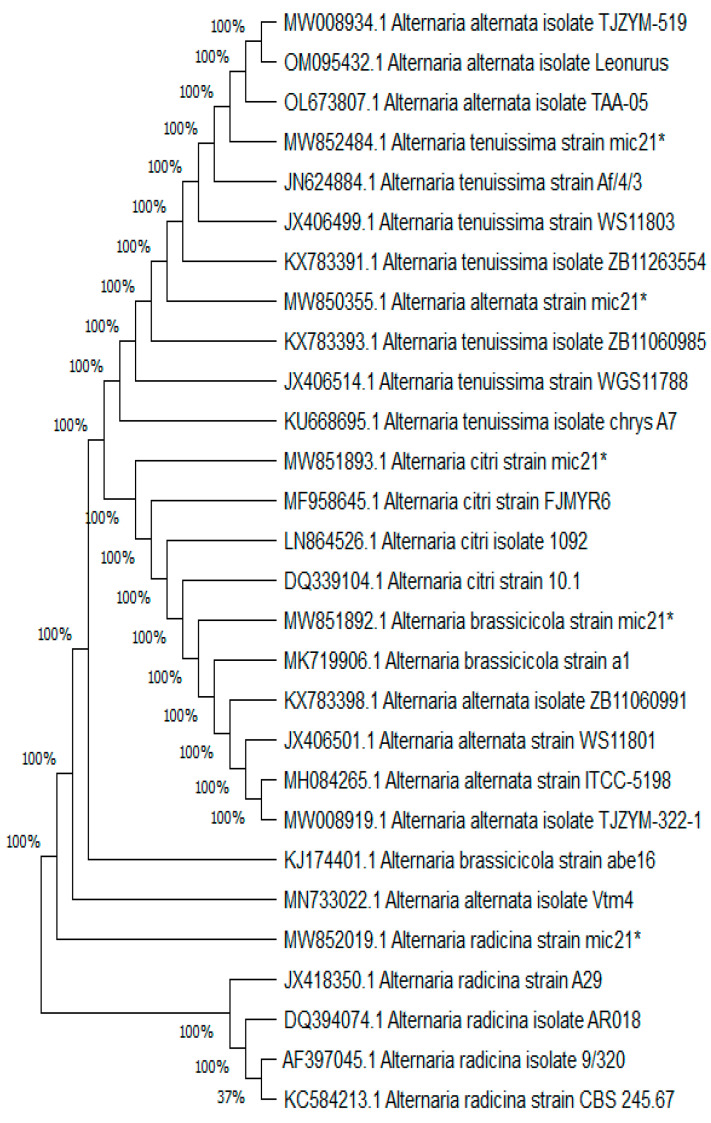
Phylogenetic tree generated from *Alternaria* species and its related species/isolates based on datasets of ITS rDNA gene sequences. There is similarity of 99–100% among species. *Alternaria* isolates with a star were recovered from tomato fruits in this study.

**Figure 4 plants-11-01168-f004:**
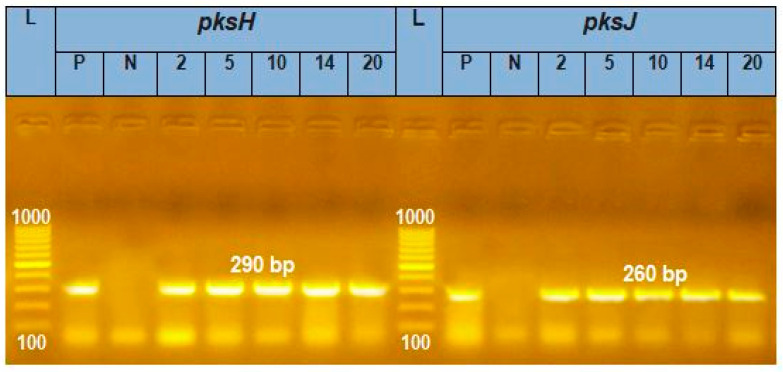
Agarose gel electrophoresis of PCR amplification of *pksH* (290 bp) and *pksJ* (260 bp) genes from *Alternaria* species: L—ladder (100–1000 bp); P—positive control consists of a segment of DNA of known size (the same size as the target amplicon, shows that the primers have attached to the DNA strand); N—negative control: a sample without DNA, but contains all essential components of the amplification reaction shows if contamination of the PCR experiment with foreign DNA has occurred; 2—*A. alternata*; 5—*A. citri*; 10—*A. brassicicola*; 14—*A. radicina*; 20—*A. tenuissima*.

**Figure 5 plants-11-01168-f005:**
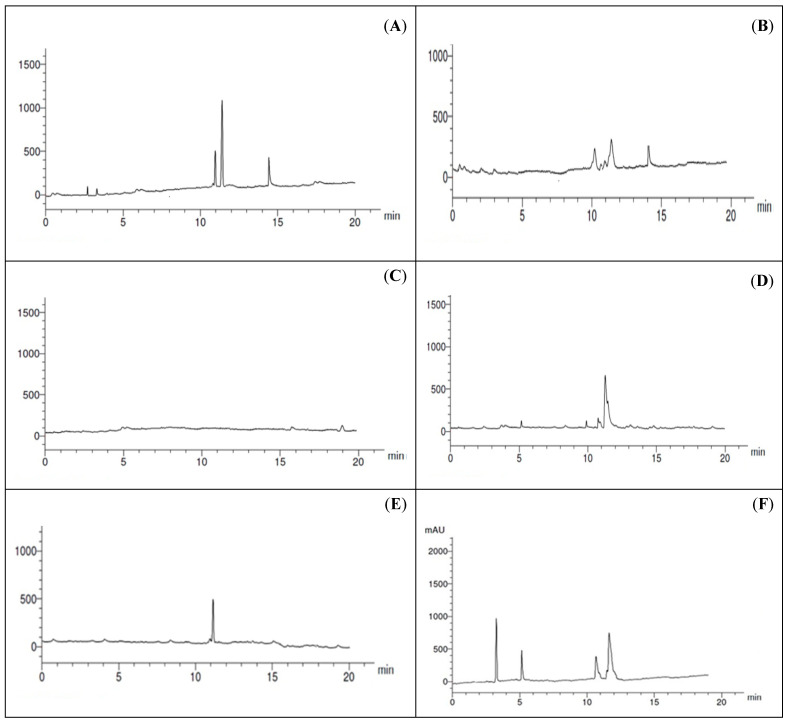
HPLC analysis of *Alternaria* culture extracts for mycotoxin detection: *A. alternata* (**A**); *A. citri* (**B**); *A. brassicicola* (**C**); *A. radicina* (**D**); *A. tenuissima* (**E**); toxin standard (**F**); AOH, AME, ALT, TA, respectively.

**Table 1 plants-11-01168-t001:** Measurement of different morphological structures of *Alternaria* species (averages of 10 conidia for each isolate).

Isolate	Species	Average Conidial Length (µm) ± SD	Average Conidial Width (µm) ± SD	L/W Ratio ± SD	Average of Transversa Septa ± SD	Average of Longitudinal Septa ± SD	Average Transversa/Longitudinal Septa ± SD
1	*A. alternata*	22.6 ± 7.0	11.4 ± 1.6	2.0 ± 0.5	3.1 ± 1.7	2.6 ± 0.8	1.2 ± 1.1
2		28.2 ± 6.74	10.4 ± 1.78	2.7 ± 0.87	3.3 ± 1.1	1.4 ± 0.66	2.4 ± 0.95
3	*A. citri*	29.4 ± 11.18	15.9 ± 3.07	1.9 ± 0.73	4.3 ± 1.25	2.5 ± 0.97	1.7 ± 0.98
4		27.1 ± 5.11	10.5 ± 1.70	2.6 ± 0.80	3.2 ± 1.01	2.4 ± 0.92	1.3 ± 0.99
5		41.4 ± 11.88	11.4 ± 2.64	3.6 ± 1.21	5.9 ± 1.45	1.4 ± 0.52	4.2 ± 1.86
6		40.3 ± 10.11	11.0 ± 2.50	3.7 ± 1.22	5.6 ± 1.11	2.0 ± 0.88	2.8 ± 0.76
7	*A. radicina*	24.4 ± 8.84	10.6 ± 3.53	2.3 ± 1.56	3.0 ± 1.49	1.6 ± 0.84	1.9 ± 1.53
8		34.8 ± 5.66	15.2 ± 1.72	2.3 ± 0.51	3.8 ± 1.40	2.5 ± 0.85	1.5 ± 1.32
9		33.2 ± 4.5	15.3 ± 1.60	2.2 ± 0.40	3.8 ± 1.20	2.0 ± 0.98	1.9 ± 1.44
10	*A. brassicicola*	30.6 ± 14.61	15.3 ± 1.43	2.0 ± 1.02	6.0 ± 2.31	3.2 ± 0.79	1.9 ± 0.72
11		32.8 ± 2.35	16.2 ± 1.68	2.0 ± 0.40	5.4 ± 0.70	2.2 ± 0.42	2.5 ± 1.07
12	*A. alternata*	31.4 ± 5.53	14.3 ± 2.88	2.2 ± 0.91	3.8 ± 1.14	1.7 ± 0.82	2.2 ± 1.38
13		21.5 ± 1.33	10.1 ± 3.34	2.1 ± 0.39	3.7 ± 1.37	1.3 ± 0.66	2.8 ± 0.67
14	*A. radicina*	31.4 ± 5.28	15.5 ± 2.03	2.0 ± 0.48	2.0 ± 0.82	3.1 ± 0.57	0.7 ± 0.28
15		28.5 ± 7.61	15.8 ± 2.33	1.8 ± 0.48	3.1 ± 1.20	1.8 ± 1.14	1.7 ± 0.88
16		24.8 ± 3.39	13.1 ± 2.54	1.9 ± 0.32	3.1 ± 0.34	1.2 ± 0.32	2.7 ± 0.60
17		29.0 ± 8.61	13.7 ± 1.94	2.1 ± 0.95	2.6 ± 0.97	1.3 ± 0.48	2.0 ± 0.84
18		34.1 ± 4.30	11.3 ± 2.35	3.0 ± 1.22	4.1 ± 1.55	1.5 ± 0.54	2.7 ± 0.63
19	*A. tenuissima*	28.7 ± 2.98	12.4 ± 1.20	2.3 ± 0.33	3.2 ± 0.42	1.5 ± 0.53	2.1 ± 1.07
20		36.0 ± 6.20	11.0 ± 0.93	3.3 ± 0.71	4.6 ± 1.65	1.5 ± 0.71	3.1 ± 1.41

**Table 2 plants-11-01168-t002:** HPLC analysis of *Alternaria* toxins (calculated by µg/mL of fungal extracts).

Mycotoxins	RT	*A. alternata*	*A. brassicicola*	*A. citri*	*A. radicina*	*A. tenuissima*
Altenuene	10	ND	ND	1.35	ND	ND
Alternariol	11	5.12	ND	ND	0.89	4.56
Alternariol methyl ether	11.5	9.85	ND	5.12	6.65	ND
Tenuazonic acid	14.8	4.12	ND	1.51	ND	ND

ND = non-detectable; RT = retention time

## Data Availability

The datasets generated and/or analyzed during the current study are available from the corresponding author upon reasonable request.

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
