# Peer review of "Morphological and Molecular Characterization of Some Alternaria Species Isolated from Tomato Fruits Concerning Mycotoxin Production and Polyketide Synthase Genes"

_plants, 2022, doi:10.3390/plants11091168_

Round 1
Reviewer 1 Report
The paper is well structured and the topic could be of interest, but some improvements are needed. Overall, as a reader I don't understand what was the purpose of the work, why the authors have chosen to make this three groups of analysis and what are the conclusions. here are my detailed comments chapter by chapter:
Introduction:
generally well constructed, but, in my opinion, some additional information on Alternaria mycotoxins should be added. When you state that some secondary metabolites are classified as mycotoxins and you choose to analyse biosynthesis of four of them, it would be good to shortly describe their toxic effects (i.e. gastrotoxic, hepatotoxic or similar) and explain that these toxins, at the best of our current knowledge, are not highly toxic.
Material and methods:
the title of sub-chapter 2.7 does not correspond to the content, it is not HPLC analysis but toxin production and purification for HPLC analysis
Results are correctly reported
Discussion:
generally in the discussion the authors are reporting results of other studies without relating those data with those obtained in this research. You should comment your data in relation to those already known. Furthermore, from the discussion it is not clear why the authors decided to analyse pksH and pksJ and what is the meaning of the results. What information is possible to obtain from these results? Why you discuss pksI when it was not analysed in your research. Is there any correlation between analyzed pks and toxin production? Regarding the mycotoxin data, are they correlated/consistant with those present in literature? How do you explain that pksH and pks J are produced by all strains and some strains produce no mycotoxins?
In my opinion, the data presented are interesting, but the paper in the actual form is not ready for publication.
Author Response
Dear, Editor
Please see the attachment. Authors responses to reviewer 1 comments yellow highlighted

Reviewer 2 Report
Line 90, 105, 107, 110, 112 – the scientific name of species has been written in italic format, please correct it in all manuscript
Line 100, 113 – Insert extra space
Line 127 - 130 – include space between the units and number
Line 129 – Insert extra space
Line 154 – Attention the units written mL not ml
Line 201 – Remove extra space
Author Response
Dear, Editor
Please see the attachment. Authors responses to reviewer 2 comments green highlighted

Reviewer 3 Report
I find data presented in the manuscript potentially interesting and important, but in my opinion this manuscript needs improving of the entire text. More general approach in introducing the subject of the study and the purpose of the work should be included, in addition to the description of the results obtained in the previous studies. Also, it is confusing when you provide the data such as line 47-49 “The implementation of the A. alternata genetic diversity by the ISSR marker could make it easy to identify appropriate and efficient strategies to decrease the fungal and mycotoxin contamination of human food.”, but in the same time you are not using this approach in your study. You indicate that the main step in the biosynthesis of mycotoxins and other fungal metabolites is polyketide synthases (pks) and that PksJ and pksH genes are essential for enzymes secretion associated with the production of toxins by Alternaria. However, it is unclear the purpose of performing only amplification of the short segment of this sequence and presenting PCR products on agarose gel. Did you not expect all your strains to have these genes? If you perhaps wonted to measure expression of polyketide synthases in different isolates, then your method is not appropriate for this. Furthermore, you had 20 samples in your study. Why only 20 samples, was there no more symptomatic fruits? Or you randomly selected only 20, but there was more? I think that it is quite important to indicate is this pathogen widespread in your region or only sporadically found it at some markets that you inspected. Related to that, I also think that you should more thoroughly discus your results and the implications of your findings.
In presenting your results you should avoid methodological data which should be included in Material and methods and avoid repeating. I find your data on molecular characterization of Alternaria species not clearly presented as well as not sufficiently discussed. For example, how did you obtained data presented in Fig2B? In the Fig3 Neighbor-joining phylogenetic tree is presented, while different tree inferring method is indicated in Material and Methods. More importantly, in Fig3 A. brassicicola strain from your study is not grouped with other stains of this species. There is also no clear clustering of different species based on analyzed sequences.
For the title of the manuscript “Morphological and molecular characterization of some Alternaria species isolated from tomato fruits in related to mycotoxin production and its corresponding pks genes” I suggest English language and style check and avoidance of abbreviations. However, justification for emphasis of polyketide synthases genes in the title should be included in the manuscript as indicated above.
minor issues:
Line 32 “Based on morphological and phylogenetic issues” - replace the term “issues” with more appropriate term such as characteristics
Line 70-72 “In contrast, eight strains of Alternaria, implicated Infectoriae section, produced a few amounts that could not form mycotoxins.” – please rephrase this sentence
Line 92 “in sanitized polyethylene pages” - is the term “pages” correct term for what you used?
I suggest merging section 2.2. and 2.3.
In section “2.4. Molecular characterization of Alternaria species & DNA extraction” only DNA extraction is described, so please correct the title of this section. Also, it is not clear did you used Qiagen kit (QIAamp DNeasy Plant Mini kit) as indicated in first sentence or CTAB method described below.
Line 174 “These species were A. alternata, A. brassicicola, A. citri, A. radicina and A. tenuissima.” – species name should be in italic, please correct this throughout the manuscript
Author Response
Dear, Editor
Please see the attachment. Authors responses to reviewer 3 comments light blue highlighted

Round 2
Reviewer 1 Report
The authors have improved the introduction chapter, now it is better connected with the presented research and introduces a reader to the topic in more appropriate way. Major improvements are done in discussion chapter, now it is better supported by the results. Even if the research is not particularly original, I think that the topic and presented results could be of interest for scientific community.
Author Response
Dear, Reviewer
I would like to thank you for your useful comments that help us to improve our manuscript. Please see the attachment
Regards
Dr. Amany

Reviewer 3 Report
Although authors made an effort to improve the manuscript, I still have some significant objection.
Analysis of PksJ and pksH genes
If I understood correctly, your hypothesis was that some species or isolates may not have one or both of pksJ and pksH genes. Please include this issue in the Introduction accompanied with appropriate references that were basis for making the hypothesis that these genes could be missing in the genome of Alternaria species and/or specific isolates resulting in incapability for toxins biosynthesis. Was your hypothesis that these genes were lost during evolution or possible never acquired? Please provide the references to support your hypothesis. Also, rephrase your aim Lines 99-101 “This article aimed to investigate some Alternaria species which cause the blackspot illness of tomato fruits and the ability of these species to produce various types of mycotoxins in addition to the pks genes implicated in the Alternaria toxins biosynthesis.” so that is completely clear what was investigated in relation to pks genes. Formulated as it is, I would expect some more detailed analysis of structure and/or function of these genes.
Furthermore, please explain why you chose to analyses only 290 bp of pksH and 260 bp of pksJ and way you used primers from Garganese et al. 2016 (reference 24 in your manuscript) that were designed for real time quantitative PCR (qPCR)?
“Response 4. Data in Fig. 2B represented the similarity between the sequences of the 5 Alternaria species under study. However, Fig. 3 represented the similarity between species under study and related species in the gene bank. A. brsssicicola is not grouped with other strains of species probably due to unequal of sequence length. “
In my opinion, explanation on methods of sequence analysis and phylogenetic tree construction have to be improved. How data in Fig. 2B were constructed and how you obtained those in Fig 3? Are values on the branches represent sequence similarity or bootstrap values? Also, in caption of Fig3 you indicate Neighbor-joining, such data is not provided for Fig. 2B and in Material and Methods you indicate than Maximum Likelihood method was used. If you used two different approaches for phylogenetic analysis you need to provide explanation for choosing these different approaches and constructing two different phylogenetic threes based on one analysed locus as well as to compare the obtained results, especially if it is not fully compliant (no clear grouping of isolates belonging to different species is visible at Fig. 3 unlike at Fig. 2B.).
DNA extraction
You responded that you used QIAamp DNeasy Plant Mini kit for DNA extraction. If so, way you describe in detail CTAB protocol in your Material and methods, section 2.3.
Subsection title: 2.4. Polymerase chain reaction, amplification of 5.8S rDNA by ITS1 and ITS4
Please recheck accuracy in the construction of this subsection title. What was amplified and way you needed to indicate name of primers used in title?
Sequencing methodology
In my opinion, you need to add more details regarding sequencing methodology, for example did you used bi-directional sequencing (along with details on sequence assembling and editing) or not, are the same primers used for sequencing as for PCR amplification or you used different primers…?
Lines 149-152 “ITS sequences revealed in our investigation, as well as validated sequences of all species discovered within Alternaria (taxid:5598), were phylogenetically examined to determine the taxonomic status of the isolates using recent molecular classification criteria [8].” – please add some more specificity on used molecular classification criteria in your study in relation to your samples and conducted analyses.
The reaction mixture for amplification of pksH and pksJ genes should be included in section 2.5.
Results:
3.2. Molecular characterization of Alternaria species using ITS rDNA gene sequencing
I find this section still not written correctly.
Five Alternaria species recovered from tomatoes were selected for molecular identification using ITS1 and ITS4 rDNA gene sequencing. – If you used bi-directional sequencing this should be described in Material and Methods as indicated above and in Results you just refer to the sequence itself which you analyzed, not primers used.
Alternaria species were designated as A. alternata, A. brassicicola, A. citri, A. radicina and A. tenuissima. – based on what? Is this designation based on morphological or molecular data? For example, BLAST for ACCESSION MW850355 (designated as Alternaria alternata strain mic21) showed more than 99% similarly with other Alternaria alternate isolates, but also with some isolates of Alternaria tenuissima and Alternaria solani. Furthermore, BLAST for ACCESSION MW851892 (designated as Alternaria brassicicola strain mic21) showed more than 99% similarly with other Alternaria species as well. It has to be clarified how you designated your isolates? And how you explain this conflicting results? Is one analysed locus (ITS sequence) enough informative to provide non-conflicting results (in my opinion Fig3 supports conclusion that it is not)? Or you might need to use multigene analysis for this purpose?
NCBI-BLAST was used to examine the obtained rDNA sequences. – is “examine” appropriate term that you wanted to use?
The morphological identification was validated by a BLAST search of the ITS rDNA sequences. – please clarify this as indicated above.
The BLAST data validated the Alternaria species variability. – please add additional explanation for this.
The three branches of the evolutionary tree of the five Alternaria species are studied in this article – this is quiet unusual description of what was analyzed in your study.
Minor issues:
Lines 45-46 “A. alternata contaminated human food and livestock by secretion of many mycotoxins.” – If you mean this in general than check grammar and use different tense. Also, add appropriate reference.
Lines 48-51 “Alternaria morphology was related to A. limoniasperae with dissimilar in conidial septa and secondary conidiophores. A phylogenetic analysis created by several sequences datasets including ITS and EndoPGgenes showed that Alternaria belongs to the Alternaria alternata group. – I do understand what you mean, but it is still not written completely correctly. Expressions “Alternaria morphology was related” and “showed that Alternaria belongs to” would mean that it applies to the whole genus Alternaria, which is not true and instead you wanted to refer to isolates analyzed by Aung et al. [14]. Therefore, please rephrase this.
Lines 48-49 “Alternaria morphology was related to A. limoniasperae with dissimilar in conidial septa and secondary conidiophores.” – in addition to the above, please check sentence structure and grammar.
Although it is generally well known, when you first mention ITS you should provide full name and description. The same is for other genes (for example EndoPGgenes) and other abbreviations used.
Lines 82-83 “…gene sequencing was used to identify altenuene, alternariol, alternariol methyl ether, and tenuazonic acid”. – please clarify how gene sequencing was used to identify mycotoxins.
Line 91 “Fungal polyketides are secreted by multi-domains.2 – please rephrase and expand this sentence to make it clearer.
Lines 110-113 “Potato dextrose agar (PDA) medium was employed for the isolation of fungi which contained 200.0 g potato; 20.0 g dextrose; 15.0 g agar (Merck, Germany) plus chloramphenicol (0.05 g/L) as a bacteriostatic agent was used for the isolation of fungi.” - please check sentence structure and grammar.
Line 118 “At 25°C, the cultures were stored for 7 days” – I think it should be: The cultures were stored for 7 days at 25°C”, same as in line 168 “At 25°C, the cultures were cultured for 15 days.”
Line 124 “Alternaria cultures grown on the PDA at 25°C to 7 days..” – should it be “for 7 days”?
Lines 145-146 “The gel was colored with ethidium bromide and experiential in a gel documentation system” – please recheck the appropriateness of the terms “colored” and “experiential”
Lines 144-146 “The PCR yields were electrophoresed (for 1 h at 80 V) in 1.0% agarose gel in Tris-borate-EDTA buffer at pH 8. The PCR yields were sequenced by Elim Bio pharmaceuticals Inc. (A biotechnology company in Hayward, California, USA)” - please check sentences structure and grammar. Also, please recheck the appropriateness of the term “PCR yields” in these sentences.
Lines 164-165 “PCR outputs were visualized below UV light in 1.0% agarose gel colored by ethidium bromide.” - please recheck the appropriateness of the terms “PCR outputs”, “below” and “colored by” in this sentence.
Lines 161-162 “PCR contained 35 cycles..” please recheck the appropriateness of the term “contained”
The reaction mixture for amplification of pksH and pksJ genes should be included in section 2.5.
Author Response
Dear, Reviewer
I would like to thank you for your useful comments that help us to improve our manuscript. The authors responses to reviewer 3 comments were green highlighted. Please see the attachment
Regards
Dr. Amany

Round 3
Reviewer 3 Report
“The phylogenetic detection of the five Alternaria species was clustered into three branches.” – I suggest removing this sentence from Abstract
“The amplified segments of the pksH and pksJ genes were 290 and 260 bp, respectively.”– I suggest removing this sentence from Abstract
The reference number [34] that you indicate to be basis for making the hypothesis on pksJ and pksH genes is in my opinion low quality article not providing any deeper explanation on this issue.
Line 106 - bp does not equal molecular weight
Lines 108-111 “This article aimed to investigate the ability of some Alternaria species which cause the black-spot illness of tomato fruits to produce various types of mycotoxins related to the amplification of pksH and pksJ genes that were involved in the biosynthesis of Alternaria toxins.” - I apologize if I was not clear, but when I suggested to rephrase your aim regarding pksH and pksJ genes, I did not mean to omit other parts of your work from the aim. Please include everything else in your aim, along with this improved and more precise explanation regarding pksH and pksJ genes.
2.4. Polymerase chain reaction, amplification of 5.8S rDNA, and gene sequencing – please correct the name of subsection
Lines 162 -164 “The reaction was made in a quantity of 25 µL containing 6 µL DNA templates, 12.5 µL master mix, and 1 µL (20 p mol) of each primer.” – no need to reaped this in the form of the table. I suggest removing the newly added Table 1
You indicate that values on the branches represent sequence similarity? Are you sure in this in both Fig. 2B and Fig. 3.? Please include explanation regarding this in the figure caption.
Also, I suggest removing description “Variation of selected Alternaria species based on ITS rDNA gene sequences (similarity of 97-98% among species)” from Fig2B and include all needed information in figure caption.
Add definition of what was used for positive and negative control in Fig2A and Fig4. You can add this in figure captions or indicate it in Material and Methods
You still state in Material and methods that you used the Maximum Likelihood method (line 183) but present Neighbor-joining phylogenetic tree in your figures. I understand that you are a beginner in this type of analysis, so I suggest you read some basic literature on phylogenetic analysis. There is a lots of books and articles available to be used as starting point. Overall, if you used Mega software check which option was selected and unify this in Material and Methods and in Results.
If you used molecular classification criteria from reference [8], why you did not include more strains from that study in your phylogeny?
Author Response
Dear, Reviewer 3
On behalf of all authors I would like to thank you very much for your valuable comments, suggestions, your kind help and support that led to the improvement of our paper. Please see the attachment.
Regards
